# Pile of Law: Learning Responsible Data Filtering from the Law and a 256GB Open-Source Legal Dataset

**Peter Henderson,**\* **Mark S. Krass,**\* **Lucia Zheng, Neel Guha**
**Christopher D. Manning, Dan Jurafsky, Daniel E. Ho**
Stanford University

## Abstract

One concern with the rise of large language models lies with their potential for significant harm, particularly from pretraining on biased, obscene, copyrighted, and private information. Emerging ethical approaches have attempted to filter pretraining material, but such approaches have been ad hoc and failed to take context into account. We offer an approach to filtering grounded in law, which has directly addressed the tradeoffs in filtering material. First, we gather and make available the Pile of Law, a ~256GB (and growing) dataset of open-source English-language legal and administrative data, covering court opinions, contracts, administrative rules, and legislative records. Pretraining on the Pile of Law may help with legal tasks that have the promise to improve access to justice. Second, we distill the legal norms that governments have developed to constrain the inclusion of toxic or private content into actionable lessons for researchers and discuss how our dataset reflects these norms. Third, we show how the Pile of Law offers researchers the opportunity to learn such filtering rules directly from the data, providing an exciting new research direction in model-based processing.

*Warning*: this paper contains quotations that may be offensive or upsetting.

## 1 Introduction

The presence of private and toxic content in the most popular corpora for pretraining large language models is a well-known problem [11, 58]. But what to do about it is largely a matter of researcher discretion. Some teams implement extensive processes for filtering content deemed toxic or private; others train on data in virtually unmodified form. Resolving all of the difficulties and nuances of content filtering can be challenging, potentially explaining why content filtering has been so uneven.

It is practically difficult to perform reliable and transparent filtering at scale. That is partially because undesirable content is deeply contextual. For example, whether the inclusion of a racial epithet in a dataset is toxic may depend on factors such as the identity of the speaker and the expectations of the readers [43, 114]. Likewise, the existence of privacy violations may depend in part on the extent to which a speaker expected a fact to be widely shared at the time it was expressed [20, 6-7]. And privacy expectations may vary widely across countries [10].

Any filtering process involves complex trade-offs. Filtering for toxicity may have unexpected effects on representation in datasets or the bias of downstream outputs [40, 48, 6]. And filtering too widely for privacy may harm important downstream applications, as when the Census Bureau's adoption of differential privacy led to errors in redistricting U.S. Congressional districts [102].

Yet researchers are not the first to balance the merits of open-source transparency with potential harms: legal and administrative actors have expended significant resources and process in developing

---

\*Equal contribution.

36th Conference on Neural Information Processing Systems (NeurIPS 2022) Track on Datasets and Benchmarks.

standards to strike this exact balance. In this work, we suggest that researchers can look to these long-developed (and debated) standards to help ground content filtering mechanisms for large language model training.

This paper makes three contributions. First, we curate and open-source a ~256GB (and growing) dataset of legal and administrative data, which we call Pile of Law, which can be used for assessing norms on data sanitization across legal and administrative settings. This dataset can be an exploratory tool for evaluating different mechanisms for "doing the data work" [104]. And we note that pretraining on the Pile of Law may help with challenging legal tasks that have the potential to improve access to justice [16]. Second, we catalog how government has, though extensive legislation, regulation, and litigation, developed standards for handling the trade-offs between privacy and offensive content on the one hand and transparency, access, and completeness on the other. We suggest actionable insights for researchers based on these legal and administrative norms.[2] Third, we demonstrate how implicit sanitization rules can be learned from the Pile of Law, providing a path forward for researchers to develop more nuanced filtering mechanisms. We also demonstrate shortcomings in alignment for current sanitization techniques, providing exciting new directions for research.

## 2  Pile of Law

We curate a ~256GB (and growing) dataset of legal and administrative text.[3] The utility of this data is twofold: (1) to aggregate legal and administrative data sources that demonstrate different norms and legal standards for data filtering; (2) to collect a dataset that can be used in the future for pretraining legal-domain language models, a key direction in access-to-justice initiatives [16]. A number of prior works have pretrained smaller models on smaller subsets of legal data, including private data that is subject to restrictive licenses [30, 129]. None of these have conducted an analysis of the legal data itself—and none have curated an open-source, legal-focused pre-training dataset at this scale.

Through extensive efforts, we compile data from 35 data sources, including legal analyses, court opinions and filings, government agency publications, contracts, statutes, regulations, casebooks, and more. Others have aggregated smaller subsets of legal data, such the EuroParl datasets which gather European Parliamentary debates [64, 57]. We have included some of these as subsets of Pile of Law when relevant and plan to continue adding material to the Pile of Law over time, further increasing its utility to the community.

We characterize the dataset in detail in Appendix E. All of the content is already entirely public and mostly available under permissive licenses, but has not previously been compiled at scale for research purposes.[4] Each of these data sources carries with it an implicit filtering mechanism formed under relevant legal standards of privacy and toxicity, which we discuss throughout subsequent sections and in the Appendix. While the underlying data in Pile of Law is already public record and has implicit filters, we recognize that it may contain sensitive material that has escaped administrative scrutiny. We discuss the ethics of our work and our proposed mechanisms for content removal in Appendix A.

This dataset has obvious utility for pretraining legal-domain foundation models, particularly since, unlike other pretraining data, all material is under open licenses. Though not central to our work, we demonstrate this potential by training an initial BERT-large equivalent model on Pile of Law, yielding comparable results to highly context specific (but smaller) models (see Appendix F for full results). Recent research has shown in legal contexts that pretraining smaller models on highly in-domain data may be better than large models on big data [129, 29]. But in theory, there should be generalizable knowledge and skills that can be learned by training across more diverse sources of data. A well-crafted pretraining procedure that instills analogical reasoning abilities, for example, should transfer across domains. Our dataset is large and diverse enough (covering distinct areas of law like criminal law, contracts, and administrative law) to test this hypothesis in the legal domain, where our initial models can form a baseline.

---

[2]Note: while we discuss a number of privacy and toxicity standards, due to the expertise of the authors and the availability of data, this work focuses on the U.S. legal system. We address this and other limitations in Appendix B.

[3]https://huggingface.co/datasets/pile-of-law/pile-of-law.

[4]See Appendix G for a discussion of copyright and licensing in the dataset.

Table 1: Filters Applied in Major Pre-Training Papers

| | PSI | Deduplication | Toxic Content | Quality |
|---|---|---|---|---|
| **CCNet** [125] | No | MinHash (pages) | No | No |
| **C4** [98] | No | Unknown (3-sentence spans) | Word list | Minimum word counts, presence of curly brackets, 'lorem ipsum', etc. |
| **GPT-3** [21] | No | MinHash (pages) | No | Train classifier to distinguish CC from curated high-quality examples |
| **Gopher** [97] | No | MinHash (pages) | Google Safe-Search | Min./max. word counts, word-to-symbol ratio, share ellipses, excessive repetition; require stop words |
| **The Pile** [44] | No | MinHash (pages) | Ad-hoc source deletion | Train classifier to distinguish CC from curated high-quality examples |

## 3 What Can the Law Teach Us About Content Filtering?

When releasing internal documents concerning individuals, courts and governments have long struggled to balance transparency against the inclusion of private or offensive content. Model creators now face a similar struggle: what content to filter before pretraining a large language model on the data. In this section we survey how governments and courts have handled such content filtering and briefly discuss how Pile of Law implicitly encodes these privacy and toxicity rules. Based on these rules, we provide actionable lessons for researchers training large language models across fields, so that they can adapt similar rules as minimum standards for dataset sanitization. To be clear, we do not take the position that legal rules are optimal nor monolithic. But in many cases they result from a deliberative process that includes judges, legislators, and policymakers in contexts open to public scrutiny, so we think that the machine learning community can at minimum learn from these laws, rules, and norms to improve current ad hoc practice. In short, there is no need to reinvent law.

### 3.1 Privacy

Despite the growing focus on privacy in NLP [20], Table 1 shows that many major pre-training papers do not explicitly filter potentially sensitive information (PSI).[5] For example, [44] excludes sources due to concerns over explicit or racist content, but does not assess the prevalence of PSI, despite including web-based sources (e.g. OpenWebText) in which users may have an expectation of anonymity. Instead, pre-training papers have focused their attention on alternatives to filtering, like deduplication [62], federated learning [108, 50], differential privacy [80, 42], and other approaches [32, 74, 65, 127, 82]. But a number of recent papers have demonstrated that large generative models output memorized content [26, 111, 32, 25, 69] even with deduplication [27]. Given that many models are trained without privacy mechanisms, filtering is critical to protecting individuals, which is perhaps why research involving health data still emphasizes that approach [90, 2]. But choosing what to filter is challenging; below, we discuss how governments and courts make such decisions.

**How have governments balanced privacy against competing values?** First, we examine how several jurisdictions handle privacy filtering. Table 2 provides a brief summary.[6]

*Baseline Redactions.* Across the jurisdictions we examine, there is a baseline level of filtering. Virtually every jurisdiction in Table 2 protects the identities of minors. At minimum, juveniles must be protected by pseudonyms in public judgments, and outside of some U.S. states, juvenile criminal records are not public. No jurisdiction normally permits the publication of financial account numbers,

---

[5]We define PSI to mean information that could violate a person's privacy interests. This could include personally identifiable information, including a person's name, date of birth, or identification number. Under this definition, a document can contain some PSI (e.g. a name or the facts of a case) while excluding other PSI (e.g. date of birth). But some information that is personally identifiable is not PSI; for example, the name and office contact information of an attorney filing a court brief is identifying but not sensitive.

[6]See Appendix I for a complete version of the table, including citations.

Table 2: Availability of Identifying Information Across Administrative Settings

| Jurisdiction | Civil Cases | Criminal Cases | Juvenile Data |
| --- | --- | --- | --- |
| U.S. Federal Courts | All case details public unless sealed, except DOBs, ID/account #s. | Def. names public; DOBs, ID/account #s, addresses redacted. | Criminal records confidential. Names redacted from civil cases. |
| U.S. Admin. Agencies | Most PII omitted from public records. | - | No statute; more protection in practice. |
| German Courts | Judgments omit all identifying information. | Confidential 3-5 years after sentence completed. | No public access to criminal records. |
| Chinese Courts | Names/case details public except in certain classes of cases. | Names/case details are public as of 2016. | Juvenile criminal records are categorically exempt from disclosure. |
| Canadian Courts | Presumption of openness, except specific details and rare sealed cases. | Public; may be sealed after a period of good behavior. | Youth criminal records are always confidential. |

dates of birth, or identity numbers like social security numbers.[7] All of these are bright line rules directly applicable to text corpora.

*Value-system contexts.* There are also significant points of disagreement corresponding to the role of privacy in different value systems. U.S., Chinese and Canadian courts denote the names of litigants in ordinary civil cases, prioritizing public access and transparency; German courts do not. Likewise, U.S. federal courts virtually never remove criminal cases from the public record [109, p. 1233], a rule also emerging in China [75]. Canada allows most criminal records to be expunged after a period of good behavior. And in Germany, virtually all criminal records are automatically sealed after a set time, and courts have even imposed fines for publicizing a person's criminal history after expungement [22].

*Contextualized privacy.* Digging further into these rules highlights how court privacy rules account for context. In the U.S. and Canada, the public disclosure of litigants' potentially sensitive information (PSI) can be avoided by persuading a court that extenuating circumstances apply [124, 115]. To name one example, courts generally permit pseudonyms when parties allege that they have suffered a sexual assault [124, p. 57]. The chance to *seal* a case, or to file pseudonymously, suggests that even the most open judicial regime allows for censoring in exceptional cases—although the sealing and pseudonymous filing standards suffer from inconsistency and misuse [113].

Likewise, administrative agencies often employ context-aware heuristics when deciding whether to include PSI in public decisions. Although administrative courts are not generally bound by stringent privacy rules like HIPAA [110, 36], the Department of Justice exempts immigration applications from public scrutiny due to privacy concerns [87]; the same is true of Social Security Disability applications. Cases involving veterans' benefits are released pseudonymously.

*Public availability is not a limit.* In many cases, the rules for sanitizing PSI and sealing cases do not depend on whether information is already public. For example, the ban on publicly filing documents revealing dates of birth in U.S. federal courts does not depend on whether a litigant's birth date is otherwise public [122]. In cases where a court does take into account the public availability of information (e.g., sealing standards [115, 121]), contextual countervailing factors can justify keeping a case sealed.

**Implications for Pile of Law.** All of the above privacy norms mean that each subset of Pile of Law is already filtered for privacy based on legal norms in that jurisdiction. Further filtering could seek to align the whole dataset with the norms in one of the subsets prior to pretraining. Appendix E summarizes the filtering norms present in each subset of the data.

---

[7]The United States' Federal Rule of Civil Procedure 5.2 lays out exceptions when these facts are contained in judicial records properly before a federal court and for civil asset forfeiture cases.

**Lessons for researchers.** First, the law provides a number of useful heuristics that researchers could deploy to sanitize data. Detecting and redacting juvenile names, dates of birth, and account and identity numbers is virtually always appropriate across countries. Legal protections for already-public information show why sanitization may be necessary even for text collected from public-facing web pages. Second, the U.S. system appears to lean most heavily toward transparency. We suggest that researchers can use the U.S. court rules as a floor. Such privacy filtering rules would already go beyond much of current modeling practice. Third, in addition to consensus heuristics, researchers should make contextualized decisions about privacy harms. While this may seem difficult, Section 4 demonstrates how to leverage Pile of Law to learn contextualized standards to mimic legal privacy redaction mechanisms; alternatively, allowing individuals whose information appears in the training corpus to request removal may serve as another stopgap. Last, the U.S. legal rules do not extend as far as some researchers suggest is necessary. For example, [5] suggests that *all* names must be redacted to preserve privacy. This would reflect greater privacy protection than is typically afforded by U.S. law, which prioritizes public openness and transparency about court proceedings, but would be in line with German rules. These pose important value tradeoffs, and we suggest that researchers look as a starting point to the jurisdiction that aligns with such value tradeoffs for filtering other potentially sensitive content.

## 3.2 Toxicity

**How is toxic speech defined in research?** The category of 'toxic speech' is defined in multiple ways [47, 123, 4]. Some papers define toxicity as "*disrespectful* comments, including . . . identity *attacks*, profanity and *threats*," thus emphasizing the idea of intentional insult [39, 126, 128]. A broader definition would incorporate *implicit* toxicity, as when a speaker "subtly" or "unconsciously expresses a prejudiced attitude" [18, 70]. [18] cites the example of the question "But where are you from, originally?" Others would take a still broader view, suggesting that any *profanity* is toxic (in addition to hate speech and derogatory content) irrespective of speaker intent [89]. One implication of these divergent choices concerns *mentions* of toxic language, where a speaker refers to something said by another [114]. For example, if a judge writes that "Plaintiff claims that her supervisor called her '___'" (where ___ is a profane epithet), an intent-based standard typically would not deem the use of ___ 'toxic,' while an approach targeting profanity typically would.

**How have governments regulated toxic content?** Scholars have documented the role of the law in institutional racism and other forms of oppression, and legal materials from prior eras use words that would by modern standards be considered epithets [31, 13]. Today, the legal profession in most Anglophone countries strongly polices overt discriminatory epithets [38]. Overtly biased speech is prohibited for judges and lawyers in the U.S., Canada, and the U.K. by professional rules [3, 24, 119, 68]; similar norms have been put forward by the U.N. [120]. Judges and lawyers in all countries are routinely sanctioned when they use racist epithets, and most incidents occur verbally or off the bench [38, 60, 118].

Unlike overt, indecorous epithets, legal norms permit the use of speech affected by implicit bias; the incidence of such speech is well-documented [94, 100, 85, 12]. Indeed, it is sometimes encoded in the laws judges and administrative agencies enforce [23]. Furthermore, some lawyers may see themselves as professionally *obligated* to deploy stereotypes when doing so may assist their clients (e.g. immigration [88], defendants in sexual assault cases [34]).

**Implications for Pile of Law.** The adversarial legal system in many Anglophone countries creates incentives for lawyers to complain about overt racism in written materials, which would violate unambiguous professional rules. Thus, the appearance of epithets in our data is more likely to be confined to quotations, mentions, or to historical legal materials. However, text in our corpus may be toxic according to other definitions; for example, we are unable to quantify the prevalence of implicit biases or offensive stereotyping. Explicit racial, sexual, or offensive terms do appear in modern legal text, but most often in the form of a quotation than direct use. For instance, many cases revolve around evidence documenting racial or gender discrimination, and judges commonly spell out profane or explicit words from the evidentiary record [63, 45]. Finally, elected officials in our legislative transcripts are not bound by the same professional norms as attorneys. Additionally, an interesting future examination may note differences between civil law and common law systems,

examining rates of offensive content between the different legal systems and norms. We provide a per-subset examination of filtering norms in Appendix E.

**Lessons for researchers.** First, as is true of privacy, the toxicity norms prevalent in many legal systems offer a lower-bound for researchers. Researchers seeking to mimic the standards that apply in courts should sanitize intentional uses of derogatory terms from pretraining data. That said, current filters are not precise enough to handle this standard. Under the rules applicable to lawyers, filters based on simple word lists would be over-inclusive because they would capture *references* to offensive language that may be non-toxic in context. Second, the rules applied in courts suggest that generative models should portray toxic behavior explicitly in some contexts, either to serve the values of 'accuracy and precision' or to persuade readers [63, p. 7]; but as [43] argues, this view is contested.

Third, in language model pretraining, there may be reason to exceed minimum judicial standards depending on the length of content needed to contextualize references to offensive speech. Accessible language models like Roberta [78] have a maximum context window of 512 tokens. If a reference to offensive content spans the majority of these tokens, the model will simply uptake the offensive content as if it were being trained for *direct use*. As model contexts grow, it may become more reasonable for researchers to adopt judicial norms.

# 4 What Can We Learn from Legal Text?

As Section 3 shows, even jurisdictions that impose a strong presumption of transparency on legal documents often allow for contextual decisions that weigh this presumption against the potential harms caused by the inclusion of PSI on the public record. Reducing these rules to tools that can be deployed for filtering may be challenging. But Pile of Law encodes these contextual decisions already, providing a rich opportunity to learn context-aware norms directly. This section demonstrates the promise of Pile of Law for operationalizing legal norms. While not comprehensive, the experiments below demonstrate a path forward for replicating the content-filtering mechanisms of courts and governments by leveraging variation in Pile of Law. In particular, we show that: (1) Pile of Law reflects variation in privacy norms that can be leveraged to learn contextual privacy rules, such as when to redact names in potentially harmful situations; (2) Pile of Law reflects variation in toxicity norms over time and across contexts, toxicity filters fall short of handling these nuances, and researchers can learn much from building toxicity filters that can handle nuances in Pile of Law's text.

## 4.1 Learning Contextual Privacy Rules

**Case Study 1: Pseudonymity in Immigration Court.** The Board of Immigration Appeals (BIA) evaluates petitions appealing immigration decisions and sometimes publishes precedential decisions that affect future cases. Some cases include applicants' full names, while others replace them with pseudonymous initials. We demonstrate how subsets of the data can be used to learn the value judgements made in making this pseudonymity decision. We split cases into paragraphs and mask terms used to refer to the applicant. We train a distill-BERT base model [105] to predict whether the paragraph should use pseudonymity or not. This model achieves ∼80% F1 on the validation set. We then examine what types of content are more likely to trigger a pseudonymity recommendation by conducting a perturbation analysis. We use the Bias in Bios dataset [35], censored for names and pronouns. We prepend an additional sentence to each biography that indicates whether the person: (1) is seeking asylum or is a refugee; (2) experienced torture; (3) committed a non-violent criminal offense; or (4) committed a violent criminal offense. Figure 1 shows that asylum and torture sentences were more likely to trigger pseudonymity while criminal offenses were less likely. This aligns with federal regulations that prevent disclosure of information related to asylum or the Convention Against Torture (8 CFR § 208.6(a)). By contrast, federal regulations allow information disclosure when a criminal proceeding is involved (8 CFR § 208.6(d)(1)(ii)), though no regulation addresses criminal history.

Next, we fit a causal lexicon using the deep residualization method (and associated library) from Pryzant et al. [95]. We control for the year that a case was published since we found that some aspects of privacy standards have shifted year-to-year, which provides a unique opportunity to learn evolving standards of privacy. We select the top 100 most indicative terms for pseudonymity and remove those

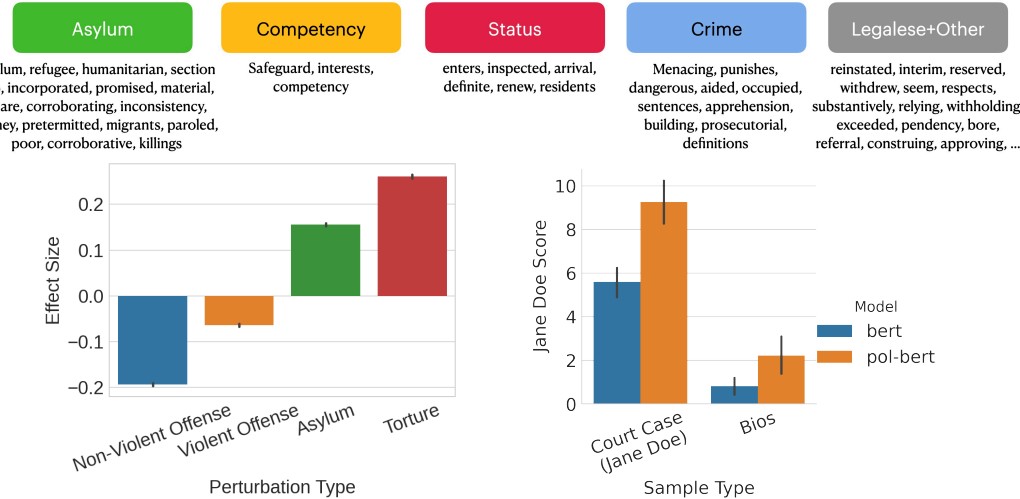

Figure 1: (Top) A causal lexicon learned for the EOIR privacy task, manually sorted by topic with contextual information. Extended version and information in Appendix H.1. (Bottom Left) A distill-bert model is more likely to predict pseudonymity for bios with an asylum or torture perturbation (effect size is difference in pseudonymity likelihood from normal bio and bio with added perturbation). (Bottom Right) Jane Doe Score is the difference in MLM score between a version of the sentence using Jane Doe and a random name. The sample sources are paragraphs using pseudonyms and Bios [35] (no pseudonyms).

where the term only showed up in one case. Then we manually examine contexts and cluster terms into categories. We found that terms most likely to be associated with pseudonymity could be largely clustered into: asylum, mental competency (a legal term used to refer to one's ability to stand trial), immigration status, and indications of a criminal proceeding. We also find that many terms associated with general legal language were included, suggested some remaining confounding and the need for more research into text-based causal attribution. These causal lexicons are seen in Figure 1.

**Case Study 2: Pseudonyms in Civil Litigation.** Next, we look to a "zero-shot" version of the experiment above in a broader setting. As noted in Section 3, litigants in U.S. courts can ask to use pseudonyms like "Jane Doe" in court documents, for example in harassment suits. To assess these requests, courts consider contextual factors like "sensitive and highly personal" subject matter, minors, or other extenuating circumstances [124]. We collect ∼ 500 paragraphs where a pseudonym ("Jane Doe" or "Jane Roe") is used from the validation part of the Court Listener Opinions data. For each sentence, we create 100 alternative sentences that replace "Jane Doe" with a name sampled using 1990 Census probabilities (using NAMES). We then compare whether each model is more likely to guess "Jane Doe" using MLM Score [103]. We repeat this process on the Bios dataset [35]. Figure 1 shows that a model trained on Pile of Law (pol-bert) ranks Jane Doe ∼ 3 points higher than a standard bert-large-uncased on true pseudonym cases. This suggests that models pre-trained on Pile of Law are more likely to encode appropriate pseudonymity norms. To be sure, pol-bert is slightly more biased for Jane Doe use overall, as is to be expected, but its performance gains persist even after accounting for this bias.

**Case Study 3: Privacy Standards in Medical Cases.** We examine *inter-source variation* between the Board of Veterans Appeals (BVA) and the Department of Labor's Employee's Compensation Appeals Board (DOL). Leading tools for data sanitization remove personal health information as defined by HIPAA [33], including dates or the name of a physician [90]. We ran [90] on all decisions by the BVA and DOL since both adjudicate the extent of applicants' disabilities, though they are not bound by HIPAA [36]. Showing the difficulty of applying sanitization tools out of domain, virtually *all* decisions included information flagged as HIPAA-protected: 99% included dates; 96% of BVA and 100% of DOL decisions included medical facility names. But the two agencies also differed. About 26% of DOL cases but just 0.36% of BVA cases included a physician name. Physician fraud is more common in worker's compensation programs like DOL's [91], but the BVA relies on the

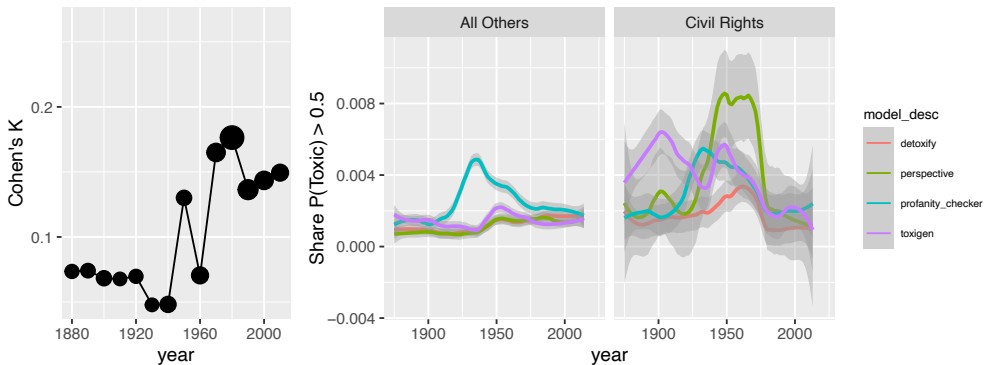

Figure 2: Inter-Model Agreement and Toxicity Over Time. Left: Cohen's $\kappa$, by 10-year bin, calculated for [130] and [59], with dot size proportional to number of examples. Right: Share of sentences assigned >50% probability of being toxic, by model, time, and topic classification [112].

testimony of VA physicians. The transparency in DOL opinions reflects the higher public interest in physician accountability.

**Lessons for researchers.** These experiments show that the Pile of Law encodes signals about privacy standards that can be learned to produce more nuanced recommendations about filtering. For example, researchers may consider whether to mimic the EOIR standard to remove names in proceedings related to minors, asylum or safety concerns. Or they may wish to learn and apply the more contextual standard that is used in general U.S. litigation, where a complex set of factors is used to justify the exclusion of names from case texts. Such contextualized filters may help ensure that generative models strike the right balance between accuracy and privacy protection, for example by accurately distinguishing benign releases of names and contact information (e.g., in response to queries about government officials) from harmful ones (sensitive circumstances where harm is plausible).

## 4.2  Calibration and Value-Alignment in Toxicity Filtering

We also identify three main insights (and challenges) from using toxicity filters on Pile of Law, setting the ground for new research using the dataset: (1) toxicity filters often disagree, creating potential issues for automated filtering; (2) toxicity filters may be value-misaligned when it comes to content that is flagged in Pile of Law; (3) toxicity scores vary highly with the length of the content, making it unclear how to handle long-document filtering.

**Case Study: Supreme Court Decisions.** Leveraging Pile of Law, we show that there are profound nuances to filtering toxic content. First, toxicity filters encode value judgements and divergent definitions of toxicity. Figure 2 shows Cohen's $\kappa$ between profanity-checker and Perspective over time for sentences in Supreme Court cases (Fig. 6 shows the same for all filters). At the sentence level, the tools' agreement rates are very low, but rise over time, indicating the challenge of handling out-of-domain data far away in time. A vivid example of this challenge is provided in Table 3: *Dred Scott* is the most notoriously racist decision in U.S. history [46], but perhaps due to the archaic language of its holding, *none* of the models is sure that it is toxic.

But civil rights cases illustrate why the disagreement is about conceptual differences, not just domain drift. Figure 2 shows that the period between 1950 and 1970 is associated with a large spike in the share of sentences deemed toxic in U.S. civil rights cases. This period was associated with the end of *de jure* segregation in the United States [117]. Many cases likely *quoted* or *mentioned* racist laws before striking them down. For instance, in Table 3, *Loving* describes a law banning interracial marriage in order to deem it illegal. Quoting this language qualifies as toxic under some but not all definitions, and as Figure 2 shows, that view is encoded in some but not all filters. Accordingly, the filters disagree as to whether *Loving*'s quote is clearly toxic. Document-level filtering could thus easily delete core civil rights cases like *Hunter* and *Loving*—while leaving in *Dred Scott*.

Finally, we note that the context window used to filter out sentences appears to dramatically influence ratings. Perspective segments data into sentences and then labels each sentence, which is the

Table 3: Toxicity Ratings of Quotes From the U.S. Supreme Court, Showing Rating Disagreement

| Case | Quote | (1) | (2) | (3) | (4) |
|------|-------|-----|-----|-----|-----|
| Hunter v. Erickson (1969) | "The majority needs no protection against discrimination." | 0.02 | 0.05 | 0.00 | 0.81 |
| Loving v. Virginia (1967) | "[I]f any white person intermarry with a colored person . . . he shall be guilty of a felony and shall be punished by confinement in the penitentiary . . . ." | 0.52 | 0.54 | 0.60 | 0.94 |
| Dred Scott v. Sandford (1857) | "A free negro of the African race whose ancestors were brought to this country and sold as slaves is not a citizen within the meaning of the Constitution." | 0.29 | 0.50 | 0.26 | 0.54 |

*Note:* Model **(1)** is profanity-check [130]; **(2)** is Perspective [59]; **(3)** is Detoxify [49]; and **(4)** is Toxigen [51] .

approach we take above. We find that by using longer span, we can *systematically decrease* the perceived toxicity of a span, even if it is obviously toxic under any definition. We take the top 5k sentences labeled as toxic by Toxigen. We then take 2 sentences before and after the toxic sentence (clamped to the boundaries of the document). We find that the toxicity score drops between **55-57%** (absolute, 95% CI) just by adding this context. While some of this change might be due to correct re-classification of mentions, we provide qualitative examples in which this is clearly untrue in Appendix Table 8.

**Lessons for researchers.** The experiments above demonstrate that, while toxicity filtering is important to align with the courts' modern lower bounds banning uses of epithets, it is not clear that existing filters are not consistent and filter out content aligned with different values. Moreover, they can arbitrarily label content as non-toxic in long-document or out-of-distribution settings, which may affect filtering mechanisms. More work is needed to create robust, value-aligned toxicity filters for pretraining and it is unclear if off-the-shelf mechanisms strike the right balance. As we have shown, the Pile of Law provides unique opportunities to develop such methods.

## 5   Conclusion

In this work we have examined how the law and legal data can inform data filtering practices that are of great importance to responsible large language model training. We provide an extensive legal dataset (the Pile of Law) and illustrate a number of exciting new research directions for future work.

## Acknowledgements

We thank SambaNova Systems for generously providing compute resources via the SambaNova Systems Dataflow-as-a-Service™ platform and the Stanford Institute for Human-Centered Artificial Intelligence for computing support. We also thank Jieru Hu for helpful discussions and Krithika Iyer for technical assistance. PH is supported by an Open Philanthropy Project AI Fellowship.

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
