# OpenReview forum: "Pile of Law: Learning Responsible Data Filtering from the Law and a 256GB Open-Source Legal Dataset"
_NeurIPS.cc/2022/Track/Datasets_and_Benchmarks — NeurIPS 2022 Datasets and Benchmarks _

### Official Review · Reviewer_zrN6 · 2022-07-22
**The timely contribution for responsible data filtering grounded in law.**

**Rating:** 7
**Confidence:** 3

**Strengths:**

- A study of the timely issue in a legally grounded way.
- The release of comprehensive large-scale English legal corpus.
- Showing a potential for a data-driven contextual text sanitization.


**Weaknesses:**

- The released corpus is mostly unstructured. For instance, in “courtlistener_opinions”, the meta data like issued date, the name of judges, the name of court, are not separated from the raw text as well as the major fields like “facts”, “claim”, and “opinion”. This may limit the usability and the uniqueness of the released dataset.
- The accompanying PoL-BERT language models do not seem to be trained optimally and their usefulness may be limited.

**Additional Feedback:**

It might be easier to understand the qualitative feature of the individual corpus if some examples are available from Appendix without necessarily downloading the large amount of data.

In Ch 4.1, line no. 225, how the ground truth is prepared to train distill-BERT?

**Clarity:**

The paper is well-written. The use of many legal terms and advanced vocabulary may decrease the readability especially for non-native English speakers but this may not be avoidable.


**Correctness:**

Their claims are solidly grounded.


**Documentation:**

Yes.

**Ethics:**

Yes.

**Relation To Prior Work:**

Yes.

**Summary And Contributions:**

The paper studies responsible data filtering practice for AI researchers. The paper conveys the legally grounded lessons by surveying how governments have developed the data filtering algorithm while balancing transparency vs privacy and toxicity. They also provide large scale comprehensive legal corpus that can be used for training legal language models and show that such models have a potential to be used as contextual data filters by performing case studies.

Overall, the paper is well written and the issue is timely. The released corpus is comprehensive and will be useful for studying legal AI. The released language model PoL-BERT does not seem to show the full potential of PoL dataset yet the focus of this paper is raising the issue and convey the legally grounded lessons with new research direction using large scale legal corpus.

---

### Official Review · Reviewer_DWPM · 2022-07-26
**A Review of Pile of Law: Learning Responsible Data Filtering from the Law and a 256GB Open-Source Legal Dataset**

**Rating:** 8
**Confidence:** 3
**Correctness:** As far as this reviewer can judge the…
**Clarity:** The paper is very well written.

**Strengths:**

* Very large dataset.

* Dataset comes from different sources.

* Can be used to answer practical questions in training large language models.

* Well documented.

**Weaknesses:**

* From a legal point of view, material from the EU does come from a different tradition than the case-law used in US and the UK. It is mentioned that "The adversarial legal system in many Anglophone countries creates incentives for lawyers to complain about overt racism in written materials". The difference between case-based and continental (or civil law) would be interesting to pursue.

**Additional Feedback:**

None.

**Documentation:**

There is extensive documentation, both in the form of appendices and in the form of the material provided online.

**Ethics:**

The authors' elaborations on the ethical dimension of their dataset are satisfactory.

**Relation To Prior Work:**

Yes.

**Summary And Contributions:**

Privacy and toxic content is a problem for training material in large language models. The paper investigates the issue by curating and using a large open source legal dataset where there data has already been filtered through legal and administrative processes. As the dataset comes from different sources (35), in different contexts (e.g., criminal vs civil cases), and in different countries, it has been filtered using different rules, as they apply in each particular case. This variegated dataset allows the authors to carry out a series of experiments leveraging the differences in the (already) applied filtering rules:

* How do different jurisdictions handle privacy filtering?
* How is toxic content handled by governments?

Also, the dataset can be used so that models can be trained in order to:

* Learn contextual privacy rules (like pseudonymisation).
* Investigate how consistent toxicity filters are ( they aren't).

The dataset may be useful to other researchers, particularly those wishing to investigate the impact in training of differentiated, vis-a-vis the filtering rules applied, of training datasets.

---

### Official Review · Reviewer_c6sX · 2022-07-28
**A massive legal text dataset**

**Rating:** 8
**Confidence:** 4
**Correctness:** they sounds correct
**Clarity:** the paper is written well

**Strengths:**

- a new dataset has been presented
- open-source dataset without license restrictions
- description of the privacy concerns
- a detailed description of data acquisition and data sources

**Weaknesses:**

- unfortunately, it contains mainly US/Canada legal domain

**Additional Feedback:**

I'm looking forward to the extension of the POL with non US/Canada legal texts

**Documentation:**

the HF dataset is described very well

**Ethics:**

all ethical consenrs are described

**Relation To Prior Work:**

it would be great to talk a little bit more about europarl datasets such as in https://paperswithcode.com/paper/europarl-st-a-multilingual-corpus-for-speech

**Summary And Contributions:**

The paper presents a dataset of open-source English-language legal and administrative data, covering court opinions, contracts, administrative rules, and legislative records.

---

### Official Review · Reviewer_RkuH · 2022-07-28
**A large dataset created with little supervision**

**Rating:** 5
**Confidence:** 3
**Clarity:** The paper is well written and easy to…

**Strengths:**

1. It is a really large dataset (256 GB). Using this as pre-training data would provide useful insights to the community, and the authors built and trained such a model.

2. The paper also shows some case studies related to the use of this dataset. This will be especially useful for researchers who are not familiar with the insights that legal documents can provide. (For example, studying anonymization or toxicity.)

**Weaknesses:**

1. The dataset can be used in the future for pretraining, but it appears it is across a very wide legal spectrum. Is this the norm in the legal domain? In healthcare, such a wide use of datasets will not lead to useful insights. For example, EHR records alone can vary based on format, structure, and term usage, and it is rarely possible to use them with a huge data dump.

2. This dataset is rather large and can explicitly violate the privacy concerns of those mentioned. I am not convinced with everything in the paper that the authors have taken appropriate measures in deanonymizing names and identifiable information (and if such a task is even possible) [2]. A public dataset that is available over the hugging face api can violate individual privacy if the documents aren’t anonymized properly or can be deanonymized easily. I understand that this was all public data and that it was available for download before with permissive licenses, but now this data is callable with a single load_dataset call, which is slightly disconcerting.

**Additional Feedback:**

I am entirely open to changing my rating if I have missed something about the privacy concerns.

**Correctness:**

There is nothing specific about correctness to comment on since this is a dataset paper with a large data dump.

**Documentation:**

This data dump has sufficient information about the sources it used. It documents the specific projects which has curated the datasets before pileoflaw in the github repository.

**Ethics:**

Significant ethical concerns: A data dump of this scale likely has significant private information. As mentioned above, I am not convinced that the curation of such a large-scale dataset is at all possible without privacy violations. [2] showed how it was possible to extract significant private information from datasets.

[2] Carlini, Nicholas, et al., "Extracting training data from large language models." 30th USENIX Security Symposium (USENIX Security 21). 2021.

**Relation To Prior Work:**

Most likely concurrent work, but HuggingFace’s BigScience initiative spent significant time on data governance research and the authors can refer to some of the principles from that work. [1]

[1] Jernite, Yacine, et al. "Data governance in the age of large-scale data-driven language technology." 2022 ACM Conference on Fairness, Accountability, and Transparency. 2022.

**Summary And Contributions:**

The paper compiles a dataset from various sources into one unified format for use in downstream research. This dataset can be used for pre-training (since it is not labelled for any specific purpose) and can provide useful insights into various social issues and data-related insights.

---

### Review · Ethics_Reviewer_DkTx · 2022-08-22

**Recommendation:** 1

**Ethics Documentation:**

Yes, the paper explicitly addresses data collection and curation practices.

**Ethics Review:**

The paper does an excellent job discussing the ethical concerns with the dataset, and suggests some useful roads ahead for other efforts to follow.  I worry that taking the legal jurisdiction's rules on privacy and the like is too minimum of a standard in many cases, but the authors note that this is a necessary condition and might not be sufficient for resolving all ethical concerns. In general, this is a useful contribution to considerations around the ethics of LLMs.

---

### Meta-Review · Area_Chair_pH68 · 2022-09-10

**Recommendation:** Accept
**Confidence:** 4

**Metareview:**

The reviews are generally positive (though somewhat short), and a large pre-training corpus for legal text will likely be useful for NLP research. One reviewer gave a reject score (5) with the following two weaknesses:

A) The dataset comes from a wide spectrum of law-related data sources, which may differ substantially and hence limit the usefulness of the dataset.

B) Privacy


I am not too worried about Point A because large language models seem to be able to learn from diverse data sources.

Regarding Point B, the separate ethics review mentions the privacy concerns as well but finds that the submission sufficiently discusses this concern and sees no serious ethical issues.

Hence overall I recommend accepting the paper.

---

### Decision · Program_Chairs · 2022-09-16

Accept